# BOAM: A Visual, Explanatory Diagnostic and Psychoeducation System Used in Collaboration with Families—Feasibility and Acceptability for Children Who Are Non-Responsive to Treatment as Usual

**DOI:** 10.3390/ijerph192214693

**Published:** 2022-11-09

**Authors:** Eva S. Potharst, Damiët Truijens, Isabelle C. M. Seegers, Julia F. Spaargaren, Francisca J. A. van Steensel, Susan M. Bögels

**Affiliations:** 1UvA Minds, Academic Outpatient (Child and Adolescent) Treatment Centre of the University of Amsterdam, Banstraat 29, 1071 JW Amsterdam, The Netherlands; 2Research Institute of Child Development and Education, University of Amsterdam, Nieuwe Achtergracht 127, 1018 WS Amsterdam, The Netherlands; 3Developmental Psychology, University of Amsterdam, Nieuwe Achtergracht 129-B, 1018 WS Amsterdam, The Netherlands

**Keywords:** youth mental health, development, child psychopathology, executive functioning, parenting stress, diagnostic system, non-responders, intervention, mechanisms, family functioning

## Abstract

Many children referred to mental health services have neurodevelopmental problems, which are not always recognized because the resulting emotional and behavioral problems dominate diagnosis and treatment. BOAM (Basic needs, Order, Autonomy and Meaning) is a new diagnostic system consisting of imaginative models that explain the complexity of symptoms and underlying neuropsychological problems in a simple way. It is designed to be used in a transparent, collaborative process with families, so that family members can better understand the nature of mental health problems, thus increasing self-knowledge and mutual understanding. In this study, the feasibility of the BOAM diagnostic trajectory and subsequent treatment informed by this trajectory are evaluated clinically in 34 children who have not responded to or relapsed after treatment as usual (TAU). Parents completed questionnaires pre-test, post-test and at a 3-month follow-up. The treatment drop-out rate was 2.9%. Post-test, parents rated the BOAM trajectory positively. The questionnaires (measuring child psychopathology, attention, executive functioning, family functioning, partner relationships and parenting stress) demonstrated sensitivity to change, and therefore, seems appropriate for a future effectiveness study. A limitation was the high percentage of missing measurements both post-test (41%) and at the follow-up (41%). The BOAM diagnostic trajectory and subsequent treatment may be a feasible alternative for children who do not respond to or relapse after TAU.

## 1. Introduction

### 1.1. Literature Review

A relatively large proportion of children and adolescents who are admitted to mental health care have neurodevelopmental problems [1,2]. A neurodevelopmental problem means that as the child grows up and develops, neuropsychological problems become visible and impact emotional, social and cognitive development. For example, children who are classified with autism spectrum disorder (ASD) or attention deficit and hyperactivity disorder (ADHD) (Hansen et al., 2018), but also children with complex post-traumatic stress disorder (PTSD), can have neurodevelopmental problems [3,4]. The neuropsychological problems that are associated with these disorders are not specific to these disorders [5,6]. That is, a systematic review of meta-analyses that was aimed at assessing the hypothesis that the C factor (cognitive dysfunction) is transdiagnostic in psychopathology showed that, indeed, underperformance across neuropsychological domains was associated with all 12 disorders that were included in the study [6]. This supported the hypothesis that the C factor is a transdiagnostic factor related to the *p* factor, which is a model that was proposed by Caspi et al. [7], suggesting that a superordinate factor of psychopathology has stronger explanatory power than disorder-specific factors. The *p* factor was not only found in adult study samples, but also in youth samples [8]. Specific symptoms, such as suicidal thoughts, non-suicidal self-injury and psychotic symptoms were shown to reflect common mental distress (*p* factor) in different youth samples [9,10]. The *p* factor was also found to correlate with neuropsychological functioning in children [8]. A study in a community sample of 895 children and adolescents aged 8 to 18 years even showed that associations between neuropsychological functioning and psychopathology were largely accounted for via a *p* factor [11]. A study on white matter tract myelin maturation showed that increased general psychopathology (*p* factor) was associated with lower rates of myelin maturation in several brain areas over time in adolescents and young adults, which suggested that impaired myelin growth in limbic association fibers may serve as a neural marker for general psychopathology [12]. The question is, however, whether mental health professionals recognize that neuropsychological problems are not only an important factor to attend to in children with ASD and ADHD, but also in other children admitted to mental health care.

Given the association between the *p* factor and neuropsychological problems in children and adolescents, it is not surprising that children with neurodevelopmental problems have a higher risk for comorbid problems [1]. The diverse and multiple symptoms of these children often make psychological treatment complex, challenging and long-lasting [13]. Additionally, parents of children with a neurodevelopmental disorder have a higher chance of developing parenting stress [14]. Parenting stress may impact the family dynamics, and subsequently interact with the child’s symptoms. In line with this, parenting stress was found to be the most important factor predicting child stress [15]. Short-term, problem-focused protocols for individual treatment, which rely on clear hypotheses on the factors that caused or maintain the problem, can be effective in, for example, children with anxiety disorders [16], but are often not sufficient in children with neurodevelopmental problems. In addition, longer and more complex psychological treatments for children with neurodevelopmental problems, in which family dynamics are included, often miss a clear focus and conceptual model; these include child symptoms, child neurodevelopmental problems, family dynamics, the interaction between child symptoms and family dynamics, and parental psychopathology (as parents may suffer from similar neurodevelopmental problems given the genetic component in these problems). Although psychological treatment can exist for different individual and systemic elements, it is important to work from a clear understanding of the cause and maintenance of the symptoms, shared between the mental health professional and the family. In this article, we present a new diagnostic system which (I) facilitates mutual collaboration between the family and the mental health professional, (II) provides a clear understanding of the causal and maintaining factors of the problems and (III) serves as a foundation from which subsequent treatment follows.

After the admission of a child or adolescent to mental health care, usually, a diagnostic process begins. The diagnostic process often starts with an inventory, description, ordering and categorization of the problem behavior, which often results in classification of the problem. Often, a classification system such as the Diagnostic and Statistical Manual of Mental Disorders (DSM) is used for this [17], although the use of such a system increases the risk for reification and stigmatization [17,18]. The transdiagnostic approach identified several other problems related to the approach of the DSM in considering a set of symptoms as a specific disorder [19]. Mental problems are not clearly separable from each other; moreover, not only biological, but also psychological and social factors play a role in the development and maintenance of and the change in mental health symptoms. Those symptoms are dimensional, which means that they are not all-or-nothing phenomena. Furthermore, a classification is not enough to make an appropriate indication for treatment or to guide the treatment process, or to offer answers about the nature of the mental health or behavior problems to the child/adolescent and parents. Therefore, the diagnostic process also aims to form explanatory hypotheses about the development and perpetuation of the complaints. Several diagnostic systems are available that can be used in this process, such as the Psychodynamic Diagnostic Manual PDM-2 [20] and clinical case formulation [21,22]. The disadvantages of these systems are that they are highly specialized [22,23], which may limit their broad application and hinder the transmission of information to the clients; this is of extra importance in child and adolescent mental health, since information needs to be accessible for children and adults with different levels of functioning. A disadvantage of the psychodiagnostic process in general is that although the family is involved in the gathering of information, the integration of the information happens behind the scenes for the family. This way of working may negatively influence the working relationship between the family and the professional, and the active engagement of the family in the treatment process and may hinder the empowerment of the child and parents by implying that the cause of the problem is too complicated for the child and parents themselves to understand.

A new diagnostic system was developed—BOAM (Basic needs, Order, Autonomy and Meaning)—which consists of visual and imaginative models that are used in transparent cooperation between the child/adolescent, parents and the mental health professional. The BOAM models can be understood by family members of different ages and levels of intelligence and language proficiency. They provide explanations on the functioning of the mind, and offer the possibility to understand the nature of and to identify the causes of the challenges the family faces. The BOAM diagnostic process is a process of learning to understand the models, learning to relate them to both the child’s functioning and parental functioning, and forming hypotheses on the causes of the problems that the child is admitted with. This process helps the family members to gain insight into themselves and each other, to understand how different problems may interact with or influence each other, to form an understanding of the origin of the problems, and to make clear what needs to be achieved, both during therapy and in the family situation, to support the child’s development and decrease problem behavior. A joint diagnostic process with the whole family offers immediate systemic intervention plus tailored psychoeducation as a basis for treatment. The most important steps in the diagnostic process and corresponding imaginative models will be described.

### 1.2. The BOAM Diagnostic Process

#### 1.2.1. Step 1 of the BOAM Diagnostic Process: Understanding Psychological Functioning Using the BOAM Basic Model

The first step is to assess which core needs of the child or adolescent (and the family members) may be under pressure. For this step, a model showing a tree is used (the BOAM basic model, see Figure 1), which is used as a metaphor for the mind, and to describe levels of psychological functioning. This model is inspired by and based upon the pyramid of Maslow [24], which depicts the hierarchy of needs of human beings. In the original pyramid, Maslow described the needs (physiological, safety, love/belonging and esteem) that must be met before a person can achieve self-actualization. The tree also shows a hierarchy of core needs, provides an overview of both basic needs and developmental needs and gives insight into needs that are mostly fulfilled, and needs that are chronically not met.

The hierarchy of needs as shown by the first BOAM basic model is summarized in the BOAM acronym: Basic needs, Order, Autonomy and Meaning. Basic needs are formulated more broadly than physiological needs (i.e., air, food, sleep, sex, clothing and shelter), which are at the bottom of Maslow’s hierarchy; they encompass both the physiological and psychological aspects of nourishment, connection, structure and predictability, attention and safety. When the conditions are right, they produce positive experiences that allow basic needs to be met. Any occurrence, however, is experienced through sensory stimuli, which need to be processed by the neurological system [25]. The processing of incoming sensory stimuli, in which meaning is given to what happens, is called ‘ordering’ in the BOAM basic model. The concept of ordering and ordering processes is partly inspired by the work of De Bruin [26]. By continuously ‘ordering’ incoming stimuli, children, from birth, recognize patterns, both regarding the physical–concrete and the psycho-social world, which makes their surroundings more predictable [27], and gives a feeling of basic security. It also builds a frame of reference, which is the sum total of what is learned (about the self, others, the world and life) and provides context and meaning to our future experiences in a mostly unconscious and automatic way. The larger and more sophisticated the frame of reference becomes, the more ‘organizing capacity’ one acquires to oversee and understand new situations quickly and automatically. New situations can be seen as ‘order load’, and must be balanced with the order capacity to be properly ordered and understood. The concept of the frame of reference is partly based on Young’s cognitive theory [28].

According to BOAM theory, when the need for ‘order’ is met, the need for autonomy arises. In this model, autonomy does not mean independence and individuality, but good executive functioning (arising from the appropriate ‘ordering’ of physical–concrete reality), good social functioning (arising from the appropriate ‘ordering’ of social–emotional reality), both based on self-regulation (also arising from appropriate ‘ordering’). Thus, autonomy includes knowing when and how to ask for help. The work of Dawson and Guare [29] on executive functions was an inspiration for realizing the connection between ordering and functioning, both executive and social–emotional. Based on good executive and social functioning, the child’s cognitive possibilities and other talents can manifest, resulting in self-actualization. Self-actualization leads to skills that allow one to be of service to other people or the world. Self-actualization and servitude fulfill the need for meaning [30]. When development is going well, children build an appropriate frame of reference and are therefore increasingly able to function on higher developmental levels.

The BOAM basic model is applicable not only to development, but also to the daily lives of people of all ages. Basic needs and the need for order must be met repeatedly throughout our lives in order to function autonomously. Through the day, we ‘rise and fall on the BOAM tree’, according to the needs that are (not) met in that given moment. If, for instance, plans on a calendar are lost and we lose oversight of our obligations for that day, stress could ensue until order is regained. During this stress, our social functioning is limited and we are unavailable to acknowledge other people’s emotions. When this model is presented to parents, they can apply it not only to their child, but also to themselves and their own (unmet) needs.

When a child (or adolescent) is admitted for mental health treatment, the first step in the BOAM diagnostic process is to reflect on the core needs of the child, and the extent to which these can be met. The extent to which a child can fulfill their need for autonomy and meaning is more visible than the extent to which a child can fulfill underlying needs, because this is expressed in the behavior of the child. For example, the child shows fulfillment of the need for autonomy by being able to show age-appropriate executive and social functioning [31,32], and fulfillment of the need for meaning by showing motivation and eagerness to learn and develop, and by showing individuality and developing a balanced identity. However, when problems are observed in the development of the higher-order needs (autonomy and meaning), this points to problems in the fulfillment of underlying needs (the basic needs and the need for order). Well-known reasons for a lack of fulfillment of basic needs are the difficult circumstances in which a child grows up, due to, for example, poverty, severe mental health problems of the parents, or relationship problems or violence in relationships with or between the parents (*qualitatively* negative experiences). These traumatic experiences can cause disordered psychosocial ordering in the frame of reference that is negatively colored, resulting in negative self-image, or a negative view of other people or the world [28]. For example, if a child is chronically bullied at school, this will influence their frame of reference so that they may interpret future ambiguous cues from classmates as hostile [33,34].

What is less well-known is that in current society, many children suffer from a lack of fulfillment of the need for order due to (1) an overload of sensory stimuli, (2) the complexity of situations that they need to function in and (3) the relatively high standards regarding executive and social functioning, and regarding educational and other achievements, that relatively young children are expected to meet. Incoming sensory stimuli can be *qualitatively* good, coming from a positive experience. At the same time, they can *quantitatively* exceed the current ‘ordering capacity’ because of overstimulation of the neurological system or by being overdemanding of the frame of reference. This can result in a feeling of being overwhelmed and overloaded at that moment. In the long term, these experiences can hinder healthy development of the frame of reference. For example, if there is a lot of predictability in the child’s physical–concrete circumstances (meaning that the child knows what is happening, and where, when, with whom, and how it is happening), more order capacity remains available for ordering of social–emotional information, leading to a better-developed frame of reference when it comes to interpreting the behavior of others. The development of the frame of reference also depends on the frequency with which successful ordering processes have taken place in the past. In our experience, this is the case, at least to some degree, in most of the children and adolescents who are admitted to our treatment center. In support of this, several studies in clinical populations of children or in children at risk for developing conduct disorder found that more than half of these children showed sensory processing difficulties [35,36,37].

#### 1.2.2. Step 2 of the BOAM Diagnostic Process: Recognizing the Disrupted Ordering Processes Using the BOAM Order Model

When the first step of the diagnostic process reveals that a child has had traumatic experiences and/or there has been a frequent lack of fulfillment of the need for order, by being overdemanding of the neurological system and/or the frame of reference, the second step is to use the second BOAM model to recognize what has gone wrong in the ordering process and see if it relates to the trauma or overdemanding (see Figure 2). This process shows the steps of the ordering process, what is needed for a successful ordering process, and the problems that can arise in these different steps of the ordering process. The ordering starts with the ongoing, moment-to-moment incoming sensory stimulation that is associated with daily activities and ends with the response to these stimuli. To respond with functional behavior, sensory stimulation from the different needs to be integrated into information; this information must be understood by linking it to the knowledge and experience that is part of the (still developing) frame of reference, and subsequently interpreted correctly, so that it is possible to respond in a meaningful way.

The different steps of in the ordering process can be unsuccessful if the ‘ordering load’ exceeds the ‘ordering capacity’. Firstly, the multiplicity of sensory stimuli can be overwhelming and overstimulating, making it more difficult to integrate the stimuli into information [33,38]. A second problem can arise in the linking of the information to the frame of reference. The information may be too complex given the level of sophistication of the frame of reference that has been developed, resulting in confusion. Lastly, the new information can be linked to a disordered frame of reference. In that case, problems can arise in the adequate interpretation of the information. According to BOAM theory, an under-developed frame of reference poses a greater risk for an all-or-nothing way of interpreting information. If ordering problems arise, perception of reality fails, and feelings and behavior will not be appropriate for the circumstances. This generates failure experiences, based on which the frame of reference will, again, not develop properly. For example, children may perceive that someone is against them if that person does not very clearly stand up for them. The child may then have the perception of reality that someone is against them much more often than is actually the case. If children come to such a negative perception of reality often, it may further form their frame of reference in a negative way. In this way, a dysfunctional, negatively colored frame of reference may also develop in the absence of clearly traumatic experiences, due to excessive demand on the neurologic system and the frame of reference. A dysfunctional frame of reference may, of course, have large consequences for the perception of reality and behavior of a child.

Using the imaginative model of the ordering process helps children, their families and therapists to understand which steps in the ordering process pose a problem for the child. It helps to form an idea about the extent to which the frame of reference of the child is age-appropriate, and to gain an understanding of the influence of crucial experiences on the frame of reference, and the influence of the frame of reference on child behavior. It may help parents to better understand the context of child negative behavior, and thus, improve parents’ capacity to empathize with the child, and to improve their understanding of what needs to be achieved to provide appropriate circumstances to meet the child’s needs.

#### 1.2.3. Step 3 of the BOAM Diagnostic Process: Recognizing the Survival Strategies and Core Needs behind Emotional and Behavioral Problems Using the BOAM Surviving Model

From the previous model, it became clear that not only trauma but also excessive demand can lead to a disordered frame of reference, resulting in ‘ordering problems’ that lead to a perception of reality in which basic security is not experienced. The third imaginative model shows how a chronic lack of basic security can result in emotional and behavior problems (see Figure 3).

The third model assumes that a child experiencing a lack of basic safety will initially exhibit ‘primary behaviors’ aimed at restoring basic security. Busy classroom conversations during lunch, for example, can put excessive demand on a child’s neurological system or frame of reference. In reaction, he or she exhibits primary behavior by, for example, retreating to the bathroom and staying there for a while, just as a traumatized child may do because they were triggered by sensory details in that situation which are linked to trauma in the frame of reference. The need behind this primary behavior is often not recognized by those involved, such as a teacher, who might encourage the child to go back to class. Similarly, children may exhibit incomprehensible behavior because they have become triggered by sensory details in the current situation, which are linked to trauma in the frame of reference. When primary behavior is misunderstood, and therefore, rejected, the child will tend to show secondary behavior, which means that the child will try to ignore their unfulfilled needs and adjust to the situation, for which the child will need to push themselves to the limit. Children have a limited capacity for self-regulation and can only sustain it for a limited time. Sooner or later, a small or larger event can trigger a tertiary behavioral response, namely a fight, flight or freeze reaction. These are often referred to as stress reactions, but are framed as survival reactions and depicted as dragons in the BOAM surviving model [39]. These reactions can have several negative consequences, such as interaction problems within and outside the family, making it even more difficult to fulfill the core needs that were already under pressure and lead to negative self-image. A vicious cycle can develop, resulting in emotional and behavioral problems in the child and misunderstanding by those around them. The environment tends to focus on the behavior, and the unmet needs become even more out of focus.

The third imaginative model not only helps us to understand the nature of the psychopathology that the child is admitted with, but also helps to decrease the shame and guilt that is often experienced in families. This is because it shows that destructive behaviors are always the result of unmet core needs, which is true for both the child and parents. Just as with the first two models, by using the third model in the diagnostic process, psychoeducation is given to families about psychological functioning in general. The problems are seen as a logical and normal consequence of the traumatic or overdemanding experiences and the resulting ordering problems of the child, and this has a destigmatizing effect.

#### 1.2.4. The BOAM Trajectory in a Broader Context

The use of visual models provides the health professional and family the possibility to go through a diagnostic phase together in close cooperation. This means that the diagnosis does not come as a ‘fait accompli’, communicated by the mental health professional, but arises as an understandable conclusion after a transparent process during the first sessions. In the diagnostic process, parents reflect not only on their child’s functioning, but also on their own psychological functioning. This creates equality, shared insights and shared language. This, in itself, constitutes a first, positive family systemic change. The outcome of the diagnostic process gives both the family and mental health professional clarity about elements that should be important in the subsequent therapeutic process. The imaginative models are, however, not only used during the diagnostic trajectory, but are used continuously during the treatment process (for example, after a crisis) to understand the cause of the situation and determine what steps should be taken to deal with the problem and prevent similar situations in the future. The treatment process is aimed at further supporting positive family dynamics, and supporting the child’s ordering process, thereby supporting the child’s broad development and ultimately decreasing child psychopathology.

The BOAM diagnostic process and subsequent treatment are in line with the transdiagnostic approach, which recognizes common processes as underlying risk or maintaining factors for problems across different classifications of disorders [19]. This means that the BOAM models can be used with different (client) populations, both in preventive and clinical settings. Perhaps the most obvious group to benefit from BOAM are children with neurodevelopmental problems. It is known that ASD, ADHD and PTSD are associated with problems with executive functions [40,41], and often with sensory problems [42], as well. However, children with other forms of psychopathology may also suffer from neurodevelopmental problems [8], which may not be recognized, and therefore, may not be attended to in mental health care. A clinical observation, using the BOAM models with children who are not (yet) diagnosed with ASD or ADHD, is that they recognized neurodevelopmental problems in themselves. A population that may be most in need of an alternative diagnostic system that informs subsequent treatment may be children or adolescents who do not respond (well) to earlier treatment or have relapsed after treatment.

#### 1.2.5. Possible Transdiagnostic Mechanisms

Transdiagnostic approaches have been fast developing [19] and seem to be especially applicable within child and adolescent mental health care, because comorbidity is even more common and developmental change is so rapid in this phase of life compared to other developmental phases [43]. Ehrenreich-May and Chu [44] gave an overview of transdiagnostic mechanisms that explain the onset and maintenance of psychopathology in children and adolescents, distinguishing intrapersonal mechanisms (such as cognitive processes) and interpersonal processes (such as family dynamics). As treatment following a BOAM diagnostic trajectory aims to both support the child’s ordering process and improve family functioning, both intrapersonal and interpersonal mechanisms may be important in explaining outcomes of the BOAM diagnostic trajectory and subsequent treatment.

By recognizing that there is an imbalance between ‘ordering load’ and ‘ordering capacity’, children and parents could learn to diminish the ordering load by decreasing overstimulating situations and by increasing structure and predictability in the life of the child. If the ‘ordering load’ exceeds the ‘ordering capacity’ less often, this may support the ‘ordering process’; these have similarities with attention processes, such as alerting and executive attention, that allow us to take in only relevant information while ignoring other information [43]. Attention dysfunction has been described as an important mechanism for youth psychopathology [43]. On the basis of the BOAM models, it is also expected that by restoring the balance between the ‘ordering load’ and ‘ordering capacity’, executive functioning may improve. Executive functioning can be described as a set of cognitive control processes which enable self-regulation and goal-directed behavior [45]. Prior research involving 292 youths aged 13 to 22 showed that poor executive functioning predicted the *p* factor [46], which is a common factor underlying both internalizing and externalizing psychopathology. Therefore, it seems useful to measure both attention problems and executive functioning problems as outcome measures and possible working mechanisms in studies evaluating the effectiveness of the BOAM diagnostic trajectory and subsequent treatment.

If a child suffers from psychopathology, ideas about what caused and maintain the difficulties often differ between family members. These different ideas, together with uncertainties about responsibility or even blame and about possible solutions, can add to the stress that the family members are already experiencing. This may magnify existing problems. The same can happen with communication in the second important social context in which the child is highly involved: the school. By undertaking the BOAM diagnostic trajectory with both the child and parents and by including the school in subsequent treatment, ideas about the causes and consequences of child psychopathology and family problems and ideas about solutions can come together. Similarly, polarizing ideas and disagreements regarding the correct ways to react to the child’s problems can arise in the family, which can cause difficulties in the relationship between the parents. However, relationship problems between parents can also be a risk factor for social–emotional problems in children [47].

If parents gain insight into the nature of the child problems and receive tools to prevent and solve difficult parenting situations (see Section 2.3 on the treatment that follows a BOAM diagnostic trajectory), this may have a positive effect on parenting stress. Parenting stress is a negative psychological reaction to parenting demands, especially when these demands are inconsistent with parent expectations and/or when parents have insufficient resources to meet the child’s demands [48]. The BOAM models can make difficult child behavior more predictable for parents, possibly decreasing inconsistencies in parent expectations, and subsequent treatment can offer resources that could help equip parents for the parenting demands. Prior research showed that parenting stress and child behavior problems have a transactional relationship over time, meaning that not only do child behavior problems elevate the risk of parenting stress, but parenting stress also elevates the risk of behavior problems [49]. Therefore, it seems useful to also include interpersonal outcome measures (family systemic problems, parent relationships and parenting stress) in studies examining the effectiveness of the BOAM diagnostic trajectory and subsequent treatment, and study whether these could be working mechanisms.

### 1.3. The Current Study

In the current study, we evaluate the feasibility and acceptability of the BOAM diagnostic models in a diagnostic trajectory, and subsequent treatment that is informed by this trajectory, in children and adolescents who were non-responsive to or relapsed after treatment as usual. As preparation for future effectiveness studies, we assessed the feasibility of questionnaires measuring the most important primary outcomes (child/adolescent internalizing and externalizing psychopathology) of the BOAM trajectory, and possible secondary outcomes and working mechanisms (attention problems, executive functioning problems, family system problems, partner relationships and parenting stress).

## 2. Methods

### 2.1. Participants

Thirty-four children/adolescents (*M_age_* = 11.5 years; *SD* = 2.6; 20 male (58.8%) and 14 female (41.2%)) and their mothers (*M_age_* = 46.9 years; *SD* = 3.4) and fathers (*M_age_* = 49.9 years; *SD* = 5.3) were admitted to the BOAM diagnostic trajectory and subsequent treatment after having had prior treatment as usual. Child/adolescent symptoms were classified according to the Diagnostic and Statistical Manual of Mental Disorders (DSM-IV or DSM-5). Twenty-two participants (64.7%) had ASD as a primary classification. Of these participants, eight (23.5%) had one or two comorbid classifications, namely ADHD (4, 11.8%), an anxiety disorder (3, 8.8%), a mood disorder (1, 2.9%), PTSD (1, 2.9%), a learning disorder (1, 2.9%), or an oppositional defiant disorder (1, 2.9%). Six children (17.6%) had ADHD as a primary classification. Two children (5.9%) had an anxiety disorder as a primary classification. One child (2.9%) was classified with a disruptive mood dysregulation disorder and an anxiety disorder. One child (2.9%) was classified with a childhood disorder not otherwise specified. Two (5.9%) children either did not receive a classification, or the classification was unknown.

Twenty-three (67.6%) children lived with both parents. Ten children (29.4%) had separated parents, where six (17.6%) lived only or mostly with their mothers, while four (11.8%) were in a co-parenting family situation. One lived with foster parents. Six children (17.6%) did not have siblings, 19 (55.9%) had one sibling, seven (20.6%) had two siblings, and two (5.9%) had four siblings. The ethnic backgrounds of the families were Dutch in 23 (67.6%) families and non-Dutch or mixed in 11 (32.4%) families. Regarding the level of education of the parents: 26 of the mothers (76.5%) were highly educated (bachelor’s or master’s degree), one (2.9%) had an associate degree, three (8.8%) had finished high school, and in four mothers (11.8%), the level of education was unknown. Of the fathers, 24 (70.6%) were highly educated, two (5.9%) had an associate degree, and in eight fathers (23.5%), the level of education was unknown.

For 23 (67.6%) of the 32 participants, the costs were covered by the municipality. In 11 (32.4%) cases, the parents paid for the trajectory by themselves.

### 2.2. Procedure

The current study was approved by the Ethics Committee of the University of Amsterdam (2015-CDE-4401 and 2017-CDE-8422). Children and their parents either admitted themselves for the BOAM trajectory or were referred to the BOAM trajectory by their general practitioner or mental health professional. If they had prior treatment as usual, to which they did not respond, or after which they relapsed, they could be included in the current study. Parents and children were asked to participate in the current study, but they could also undertake the BOAM diagnostic trajectory if they decided not to partake in the study. Written informed consent was obtained from all participants. Before the start of the BOAM diagnostic trajectory, parents were invited to complete a pre-test assessment consisting of questionnaires that could be completed online. The post-test questionnaire was conducted immediately after the training. Lastly, the parents received the follow-up assessment 3 months after the diagnostic trajectory and subsequent treatment informed by this trajectory.

### 2.3. Intervention

The BOAM diagnostic models were developed and designed by Damiët Truijens [50]. The BOAM diagnostic process and subsequent treatment informed by this diagnostic process took place between April 2016 and June 2022. The diagnostic trajectory consisted of two to three sessions using the BOAM models involving: (1) an explanation of the models by the therapist (DT), (2) a discussion of the models between the family and the therapist in light of child functioning and development and parental functioning, and (3) the forming of hypotheses about the cause of the problems that the child was admitted with. Sessions were mostly carried out with both the child/adolescent and the parents.

Subsequent treatment was informed by the insights that were taken from and hypotheses that were formed based on the diagnostic process. In the first step of the treatment, the family, with the support of the therapist, analyzed difficult situations that they had recently encountered using the models. The aims of this step were to: (1) create an understanding about what elements in specific situations added to the difficulties (while also making clear why a child has specific abilities in some situations, but not in others), (2) increase self-knowledge and mutual understanding in the child/adolescent and the parents, (3) teach the families to use the models without the therapist, and (4) give directions for subsequent treatment, depending on the cause of the ordering problems (trauma or excessive demand). Using the models in this phase often supports the parents to pinpoint which element of the models should be focused on in the treatment.

Subsequent treatment is not protocolized but is dependent on the results of the diagnostic trajectory and the needs of the family. The type of treatment can be described as a combination of parental guidance, cognitive behavioral therapy for the child and family therapy. In this sense, the treatment phase is similar to treatment as usual. Our clinical impression is that there are two important differences between treatment after a BOAM diagnostic trajectory and treatment as usual. The first is that because both the parents and children have a greater understanding of the problems after the diagnostic trajectory, they also better understand what is needed in terms of treatment, and they may be better motivated and more compliant with the current treatment. Therefore, they may benefit more from treatment that may not be very different to the treatment received earlier. The second is that the BOAM models help parents to stay more attuned to understanding the basic needs of the child (for example, safety and predictability); the child and parents learn how to meet these basic needs, instead of relying on treatment that is focused on meeting certain expectations when the basic needs are not met. The third is that that relatively more time is spent on supporting the ordering process, including in children who do not have an ASD or ADHD classification. The fourth is that parents are stimulated to also apply the models to themselves and, for example, gain a greater understanding of what they, themselves, need to be able to fulfil the needs of the child without becoming exhausted or overwhelmed. The fifth is that the visual help model is used, which is explained in more detail in the next paragraph. The help model gives support and structure in ‘crisis-situations’, by giving parents steps to follow in these situations.

Examples of subsequent strategies for ordering problems that may be caused by excessive demand may be to: (1) reduce the number of situations that are overstimulating and/or restore the balance between ordering load and ordering capacity; (2) add more structure and predictability to family life, for example, by using pictograms and agendas, making step-by-step plans for situations that often bring difficulties, or using a list of family rules with clear consequences for situations in which the rules are not followed [26]; (3) make psychosocial situations more predictable for the child by making mindmaps [26]; (4) support the child in the processing of traumatic events; and (5) use the BOAM help model (see Figure 4) in cases of child emotional or behavioral difficulties, the so-called ‘dragon-reactions’, as explained in Section 1.2.3. Step 1 entails giving the child recognition, and especially recognizing the ‘primary problem’. In this step, parents are taught to make use of techniques that are based on non-violent resistance in families, as developed by Omer [51]. In step 3, the parents can support the child by fulfilling their core needs, which may be one of the prior described steps (but focused on the specific situation that the problem arose in), or it can consist of giving the child information (for example, by putting the event in a context that the child may not have been able to recognize), or offering cognitive restructuring.

### 2.4. Measures

*Internalizing and externalizing psychopathology* and *attention problems* were assessed using the Dutch version (CBCL) [52] of the Child Behavior Checklist for parents of children aged 6 to 18 years old [53]. The parents rated the 113 items on a 3-point Likert scale, ranging from 0 (not true) to 2 (very true or often true). Higher scores indicate more problem behavior. Good psychometric properties have been shown for the Dutch version of the CBCL [52]. The scores of two broadband syndrome scales, internalizing and externalizing psychopathology and subscale attention, were used in the current study. Examples of items of each of the used (sub)scales are: ‘Feels worthless or inferior’, ‘Gets in many fights’, and ‘Can’t concentrate, can’t pay attention for long.’ The internal consistency of the subscales in the present study was satisfactory to excellent, with Crohnbach’s *α* for internalizing psychopathology ranging from 0.85 to 0.93, for externalizing psychopathology from 0.90 to 0.94, and for attention problems from 0.75 to 0.88.

*Executive functioning* was measured using the parent-report version of the Dutch version [54] of the Behavior Rating Inventory of Executive Function (BRIEF) [55] for children aged 5 to 18 years old. Parents rated the 75 items on a 3-point Likert scale, ranging from 0 (never) to 2 (often). Higher scores indicate more problems in executive functioning. An example of an item is: ‘Forgets what he/she was doing.’ The psychometric properties of the Dutch version of the BRIEF were shown to be satisfactory [54]. In the current study, the total scale was used. The internal consistency of the BRIEF in the current study was excellent, with Crohnbach’s *α* ranging from 0.92 to 0.95.

*Family systemic problems* were measured using a questionnaire that was developed especially to measure (family) systemic problems on which the BOAM diagnostic trajectory and subsequent treatment were expected to have a positive effect. Parents rated the systemic problems questionnaire on a 4-item Likert scale, ranging from 0 (completely disagree) to 3 (completely agree). Examples of the items are: ‘Between the school and me exist different views about the right approach to solving my child’s problems’, ‘My partner’s reaction to the behaviour of my child seems to worsen the situation’, and ‘My child’s problems have a negative impact on our family life’. Internal consistency in the current study was satisfactory to good, with Crohnbach’s *α* ranging from 0.74 to 0.81.

*Partner relationship* was measured using the subscale Partner Relationship of the Family Functioning Questionnaire (FFQ, in Dutch: Vragenlijst Gezinsfunctioneren voor Ouders) [56]. The FFQ aims to measure different aspects of family functioning. The subscale Partner Relationship consists of 5 items that are rated on a 4-point Likert scale, ranging from 1 (does not apply) to 4 (applies completely). An example of an item is: ‘I feel supported by my partner in taking care of the children.’ For interpretation purposes, all scores were recoded so that, just as with all other outcomes, a higher score indicated more (partner relationship) problems. The psychometric properties of the FFQ were good [56]. The internal consistency of the PSI short form in the current study was good to excellent, with Crohnbach’s *α* ranging from 0.89 to 0.95.

*Parenting stress* was assessed using the short form of the Dutch Parenting Stress Index (PSI) [57], based on the American Parenting Stress Index [58]. Parents rated each item on a 6-point Likert scale, ranging from 1 (totally disagree) to 6 (totally agree). Higher scores indicate more parenting stress. An example of an item is: ‘Parenting this child is more difficult than I thought it would be.’ The Dutch PSI possesses good reliability [57]. The internal consistency of the PSI short form in the current study was excellent, with Crohnbach’s *α* ranging from 0.95 to 0.97.

*Acceptability* of the BOAM trajectory was measured using the number of people who dropped out and a 9-item evaluation questionnaire. The questions in this questionnaire can be found in Table 1.

### 2.5. Data Analysis

All calculations and analyses were conducted using SPSS 25 software. The power calculations were conducted using G*power. To maximize the number of completed questionnaires about each child, the following procedure was followed: the parent who completed most measurements was selected to be included in the statistical analyses. If both parents completed the same number of measurements, then the average scores of both parents were included in the analyses. Of the 34 children/adolescents who were included in the analyses, we used the questionnaires of 17 (50.0%) mothers, 6 (17.6%) fathers and for 11 (32.4%) children, with the average scores of both parents. An exception was made for the means that were calculated based on the evaluation form; for this analysis, all responses were included.

Research drop-out was defined as not completing both the post-test and follow-up questionnaires. The differences between research dropouts and other participants were tested using *t*-tests and chi-square tests. Inspection of the outcome distributions (of post-test minus pre-test difference scores) indicated sufficient normality, and the skewness and kurtosis of all variables were <|2|, except for VGFO and the systemic problems questionnaire. However, no outliers (>3.29 *SD* or <−3.29 *SD*) were found for these measures.

The confidence intervals (95%) of the mean differences between the pre-test and post-test and between the pre-test and follow-up questionnaires, and the corresponding effect sizes (Cohen’s *d*’s), were calculated. Cohen’s *d* effect sizes can be interpreted as follows: 0.2—small, 0.5—medium and 0.8—large [59]. Pearson’s correlations between the outcome measures, as measured pre-test, were calculated. The difference scores of all the outcomes between the pre-test and post-test questionnaires were calculated. Pearson’s correlations between the difference scores of the outcome measures were calculated. Due to the small sample size in the current study, we did not measure whether the differences between measurements and correlations were statistically significant.

## 3. Results

### 3.1. Research Drop-Out Analysis

Research drop-out was defined as not completing both the post-test and follow-up questionnaires. Eleven (32.4%) of the participants were research dropouts. There were no differences in the pre-test variables between research dropouts and parents who were still participating in the research after the pre-test questionnaire (*p*-values between 0.626 and 0.977). There was one borderline significant difference regarding sociodemographic characteristics: mothers who were research dropouts had a somewhat lower level of education (*p* = 0.095). There were no other differences in sociodemographic characteristics (*p*-values between 0.145 and 0.780).

### 3.2. Feasibility and Acceptability of the BOAM Diagnostic Trajectory and Subsequent Treatment

Of the 34 clients who started with the BOAM trajectory, one (2.9%) decided not to finish the trajectory. The reasons the mother gave for stopping were (1) the costs (in her case, the costs were not covered by the municipality), and (2) the mother felt that too much input from her was expected regarding the treatment goals. The other clients finished the BOAM diagnostic trajectory and subsequent treatment. The BOAM diagnostic trajectory and subsequent treatment for the participants who completed the trajectory included an average of 21.1 (*SD* = 16.6) sessions (range 7–73). The one participant who dropped out attended two sessions. The results of the evaluation questionnaire can be found in Table 1.

### 3.3. Feasibility of Outcome Measures

The means and standard deviations of the outcome measures at all measurement points are displayed in Table 2. The confidence intervals and effect sizes (Cohen’s *d*’s) of the change between the pre-test and post-test and pre-test and follow-up questionnaires are presented in Table 3.

### 3.4. Power Calculation for Future Studies

The effect sizes of the primary outcome measures ranged from about 0.60 to about 0.70. Assuming a within-group effect size of 0.60, a sample size of 32 participants would be needed to reach an alpha error probability of 0.05 and power of 0.95. On the basis of the research drop-out rate of the specific population in the current study, about 54 participants would be needed to detect within-group differences. Based on the same premise, for an analysis in which a within/between-group interaction with two measurements would be analyzed, a sample size of 147 would be needed (under the assumption that the control condition would be a waitlist condition). With a similar high research drop-out rate, 245 participants would be needed.

### 3.5. Correlations between the Outcome Measures

The correlations between outcome measures are depicted in Table 4. All secondary outcome measures showed a correlation with a medium or large effect size with at least one of the two primary outcome measures. The correlations of the difference scores (post-test minus pre-test) of all the outcome measures are depicted in Table 5. The difference scores of all the secondary outcome measures showed a correlation with a medium or large effect size with at least one of the two primary outcome measures, except for partner relationship problems.

## 4. Discussion

In the current study, the BOAM diagnostic trajectory and the models that were used in this trajectory and in subsequent treatment were presented. The feasibility and acceptability of the BOAM diagnostic trajectory and subsequent treatment (with an average number of sessions of 21.6 (*SD* = 16.6)) was evaluated in a group of children/adolescents who were non-responsive to or relapsed after regular treatment as usual. We found a very low treatment drop-out rate (2.9%) and, in general, very positive results for the evaluation questionnaire. For example, on the question regarding how suitable BOAM was as a method, parents graded BOAM on a Likert scale from 1 to 7 with an average of 6.4 (*SD* = 0.9), and the extent to which they would recommend BOAM to other families with an average of 6.3 (*SD* = 1.1), also on a Likert scale from 1 to 7. The results of the current study suggest that it would be worth further studying the effectiveness of the BOAM diagnostic trajectory and subsequent treatment for children or adolescents who are non-respondent to or relapse after treatment as usual.

The feasibility of several outcome measures was evaluated. The internal consistency of all the outcome measures was acceptable-to-good for all the measurements. The confidence intervals and effect sizes of the outcome measures that were administered in the pre-test, post-test and follow-up questionnaires suggested that change over time may be found for all outcomes in a future effectiveness study, and that the questionnaires that were chosen in this feasibility study may also be suitable for an effectiveness study. The effect sizes of (within-group) change over time in this study with a small sample size were, on average, 0.67 for the change between pre-test and post-test and 0.58 for the change between pre-test and the 3-month follow-up; however, these need to be treated with caution given the small numbers in the study, the number of participants for whom the post-test and follow-up measurements were not available and the fact that we made use of parent-reported measures only. The results of this study seem to justify a larger-scale study with a control group, to demonstrate the effectiveness of the BOAM trajectory. If a randomized controlled trial with a BOAM group and a control waitlist group was be performed, a sample size of 147 study participants would be needed.

In the current feasibility study, attention problems and executive functioning were proposed as possible intrapersonal secondary outcomes and/or working mechanisms. In earlier research on the effectiveness of interventions aimed at improving attention problems and executive function problems, the treatment of choice is often either pharmacological [60] or involves explicit training of attention or executive functioning, for example, via the daily practice of mindfulness or computer tasks [61,62]. However, a meta-analysis showed that for nontypically developing children (including children with neurodevelopmental disorders or behavior problems) the explicit training of executive functions is less effective than acquiring new strategies of self-regulation [63]. BOAM may do this by offering a method that increases understanding with regard to ordering processes and the support of these ordering processes, which may also have positive effects on cognitive development.

Positive correlations with medium-to-large effect sizes between a decrease in the primary outcomes (internalizing and externalizing psychopathology) and adecrease in attention problems and executive function problems suggest that the latter variables may indeed be suitable to study as possible working mechanisms. Prior research already showed that attention problems and executive function problems may be a mechanism in the developmental path towards child/adolescent psychopathology [43,45]. In an article on the role of disordered attention in psychopathology, it is concluded that the treatment of attentional difficulties can be an important adjunct to regular therapeutic approaches [64]. A future study could test whether a decrease in attentional difficulties and executive functioning problems during or after a BOAM trajectory may be predictive of a later decrease in internalizing and/or externalizing problems.

In this study, family systemic problems, partner relationship problems and parenting stress were proposed as possible intrapersonal secondary outcomes and working mechanisms. Earlier research showed a transactional relationship between parenting stress and family functioning on the one hand, and child behavior problems on the other hand [49,65]. Therefore, it may be extra important to intervene with both child/adolescent functioning and family functioning if a child has not responded well to or relapsed after treatment as usual. The BOAM diagnostic trajectory and subsequent treatment intentionally involves both the child/adolescent and the parents (and to some degree, the school) in the process, with the aim of improving family system functioning around the ordering problems of the child, thereby improving child functioning. Positive correlations with medium-to-large effect sizes between a decrease in the primary outcomes (internalizing and externalizing psychopathology) and a decrease in family systemic problems and parenting stress suggest that the latter variables may indeed be suitable to study as possible working mechanisms. 

BOAM theory suggests that the characteristics of modern society are risk factors for a lack of fulfilment of the need for order, and thus, for the development of social–emotion or behavioral problems. The environment that a child grows up in may be qualitatively good, but quantitatively overstimulating and overcharging due to an overload of sensory stimuli, high complexity and relatively high expectations and demands in the areas of social and executive functioning, depending on the neurodevelopmental make-up of the individual child. Attuning the environment to the needs of the individual child is important, as children also need proper stimulation and rich environments [66], and may also benefit from the positive expectations of parents, for example [67]. Research has shown that overstimulation in the form of an excessive amount of time watching rapidly sequenced television in young children is associated with later attention problems [68,69]. Although the authors are not aware of research addressing the hypothesis that growing up in complex environments is a risk factor for the development of social–emotional or behavior problems, research has been conducted on the association between living in an urban environment and mental health in children. The percentage of children growing up in urban living environments has been growing in the past decades [70]. Additionally, the prevalence of mental health disorders, such as autism, depression and anxiety, has increased [71,72]. Both autism and common mental disorders have been shown to be related to urbanicity in many different countries [73,74]. A meta-analysis on rural–urban migration and mental health in Chinese migrant children showed that compared with urban children, migrant children had more mental health problems [75]. One of the explanations that the authors gave for this difference was that mobile life and the fast lifestyle may contribute to psychological discomfort in migrant children [75]. Future research could not only assess the proper amount and quality of stimulation richness in the environment and expectations for children in general, but could also look at the right attunement of these elements to the neurodevelopmental capacities of children.

The BOAM models bring awareness to the possibility that children with social–emotional or behavioral problems suffer from ordering problems that are related to being placed under excessive demand and being overstimulated. Possible overstimulation or excessive demand are often attended to under treatment as usual for children with ASD, but are often overlooked in the treatment of children with other classifications, while it has been shown that other psychological problems are also associated with sensory processing difficulties and other neuropsychological problems [11,33]. Although ASD is diagnosed on the basis of behavioral criteria, even milder forms of ASD are often seen as a lifelong sensory dysfunction and disorder of neurodevelopment, which implies neural dysfunction in the child [76]. From the BOAM perspective, similar behavioral characteristics may be seen as a combination of a neural sensitivity and a mismatched environment, and are called ordering problems. Therefore, BOAM theory offers a more hopeful perspective with regard to ordering problems and associated social and/or executive functioning problems by revealing the choice to better adjust the environment to the needs of the child and give them more time to develop in order to avoid psychological experiences of failure and its consequences for development. The models not only identify possible problems areas, but also show the possibilities and the right conditions for healthy development.

The BOAM models aim to depict universal processes that are related to psychological functioning. They are meant to be used not only in the context of mental health care, but may also have implications for prevention programs, and for educational and care environments. What may be needed is heightened awareness that, for example, activities such as school camps or extracurricular art classes may be positively stimulating for one child, but may be too unpredictable, overstimulating or overwhelming for another child, thereby undermining the feeling of basic security for this child. Knowledge about the negative effects of avoidance on anxiety seems to be widespread, which may stimulate the opinion that children learn to deal with difficult situations, such as new or overstimulating situations, by coming into contact with them. Although this may be true, on the basis of BOAM theory, enough attention should be given to finding the right balance, and to giving the child the proper support, preparation and structure in these situations. In a report on early development and learning, emphasis was not only placed on ways in which children learn new skills and competencies, but also on the importance of secure relationships in learning (and thus, not too many changes in the people involved), and the negative effect of chronic high stress on learning [66]. That means that if a child experiences high stress or does not feel secure, this should be attended to first, and this stimulation and expectations should be adjusted for this child.

The current study had several strengths and limitations. One strength was that it was possible to include the models of this new diagnostic system. Another strength was that in evaluating the feasibility of possible questionnaires that can be used in the evaluation of the effectiveness of BOAM, we did not only focus on the effects on primary outcomes (child internalizing and externalizing psychopathology), but also on secondary outcomes and possible working mechanisms. An important limitation was that on the basis of the current study, no conclusions can be drawn on the effectiveness of the BOAM diagnostic trajectory and subsequent treatment. Another limitation was the high percentage of missing measurements for both the post-test (41%) and follow-up (41%) questionnaires. In our study sample, there was a high percentage of parents with a high level of education, and of the parents who dropped out of the research, slightly more parents had a lower level of education than the parents who did complete the post-test and/or follow-up measurement. It is unknown whether the results of this study are generalizable. Additionally, we cannot rule out the possibility that parents who dropped out of the research would evaluate the BOAM trajectory less positively than the parents who did complete the post-test or follow-up measurement. Furthermore, the measurements consisted solely of questionnaires filled in by the parents. Therefore, we do not have any information on how the children/adolescents or school experienced the trajectory. Future research could use a methodologically sound design, such as a (quasi-) experimental design to evaluate the effectiveness of BOAM, differentiate between the effects of the BOAM diagnostic trajectory and subsequent treatment, and use more diverse informants or more objective measures such as observations and neuropsychological tests. This is needed to evaluate whether there was actually an improvement with respect to the intrapersonal and interpersonal factors, and whether these can be attributed to treatment that is based on BOAM diagnostics.

## 5. Conclusions

The results of this small-scale study imply that if children or adolescents are non-responsive to or relapse after treatment as usual, the BOAM diagnostic trajectory and subsequent treatment might be a feasible alternative. BOAM does not need to replace another diagnostic trajectory but can be offered additionally. The trajectory includes both the child/adolescent and the parents, so that both may benefit from the trajectory, and both child functioning and family functioning may improve. However, the effectiveness of the BOAM trajectory is still unknown and should be studied in trials using sound methodology, an adequate sample size and a control condition.

## Figures and Tables

**Figure 1 ijerph-19-14693-f001:**
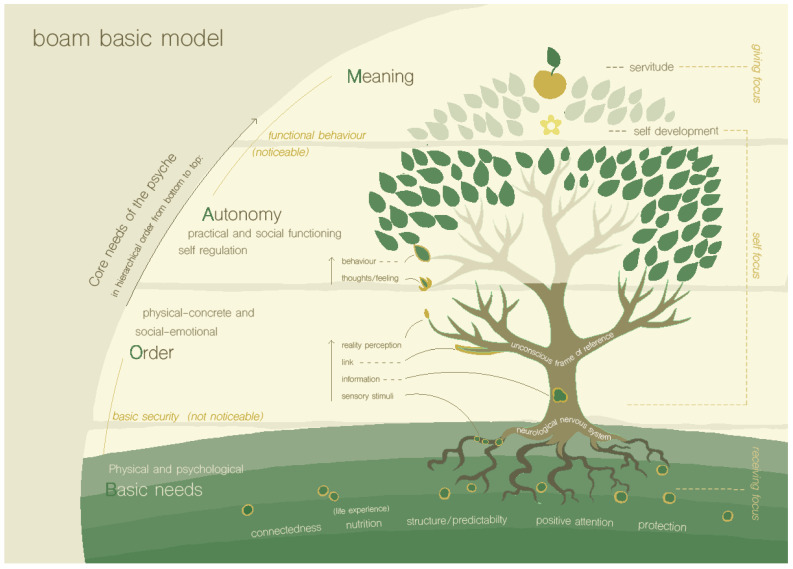
The BOAM basic model depicting psychological functioning (Truijens, in prep.).

**Figure 2 ijerph-19-14693-f002:**
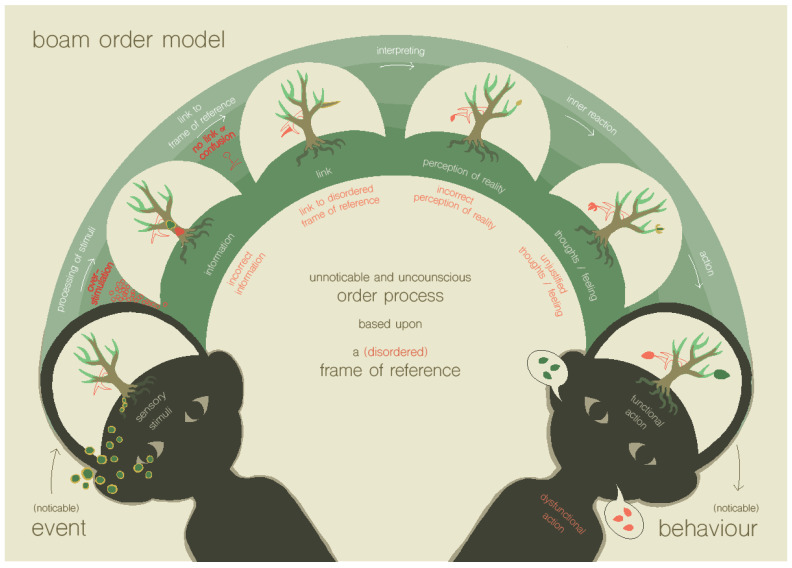
The BOAM order model, depicting the process of ordering stimuli (Truijens, in prep.).

**Figure 3 ijerph-19-14693-f003:**
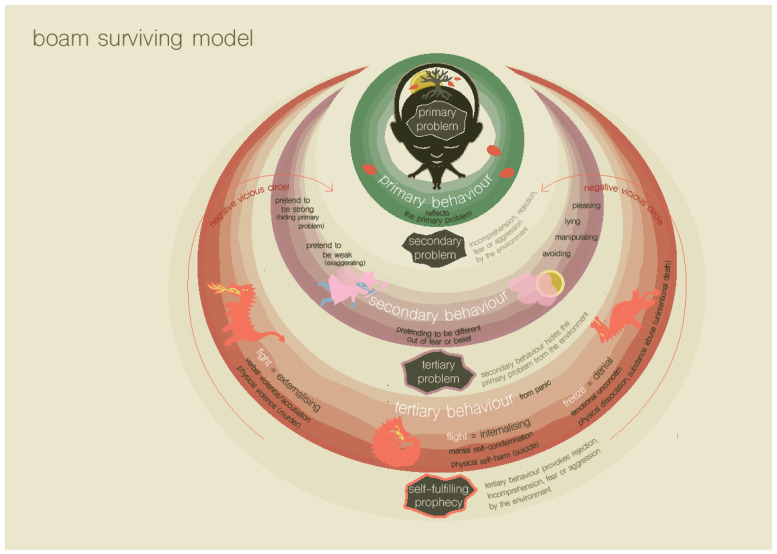
The BOAM surviving model, depicting the consequences of ‘ordering problems’ (Truijens, in prep.).

**Figure 4 ijerph-19-14693-f004:**
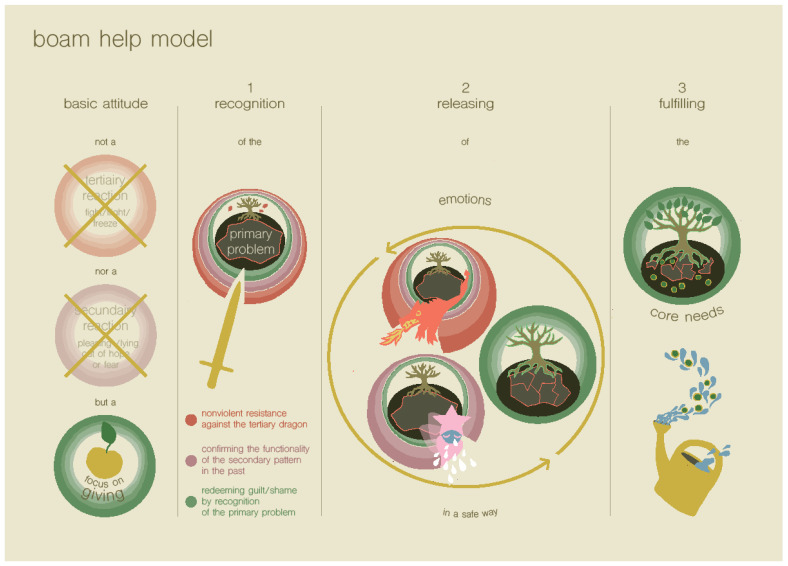
The BOAM help model, which can be used to defuse a crisis in cases of child emotional or behavioral difficulties (Truijens, in prep.).

**Table 1 ijerph-19-14693-t001:** Evaluation of the BOAM diagnostic trajectory and subsequent treatment (*n* = 31).

Question	7-Point Likert Scale
	1 = dissatisfied, 7 = very satisfied
About the improvement, I feel…	5.7 (1.3)
	1 = not suitable at all, 7 = very suitable
The method (BOAM) used was…	6.4 (.9)
	1 = strongly advise against, 7 = strongly recommend
To what extent would you recommend BOAM to other families?	6.3 (1.1)
	1 = very negative, 7 = very positive
How do you feel about what you have learned?	6.0 (1.1)
**Question**	**10-Point Likert Scale**
	1 = not at all, 10 = very suitable
In how far was BOAM suitable for the reason of referral?	8.5 (1.5)
	1 = not at all, 10 very much
How content were you about the therapist?	9.2 (1.1)
To what extent can you use what you have learned in daily life?	8.0 (1.2)
To what extend did your self-knowledge increase?	8.2 (1.1)
To what extend did your confidence in the future increase?	7.7 (1.2)

Data are presented as means (standard deviation).

**Table 2 ijerph-19-14693-t002:** Means and standard deviations of sum scores of all outcomes for all measurements; the BOAM diagnostic process and subsequent treatment took place between pre-test and post-test questionnaires.

Outcome Variable	Pre-Test		Post-Test		3-Month Follow-Up	
	*n*	*M (SD)*	*n*	*M (SD)*	*n*	*M (SD)*
Primary outcomes						
Internalizing psychopathology	34	18.1 (9.1)	20	13.6 (10.1)	20	15.0 (10.8)
Externalizing psychopathology	34	17.9 (10.0)	20	11.6 (6.8)	20	13.0 (8.7)
Secondary outcomes and mechanisms						
Attention problems	34	9.6 (3.6)	20	8.0 (3.3)	20	7.5 (3.6)
Executive function problems	33	161.7 (18.6)	18	149.9 (19.6)	18	152.6 (22.9)
Family systemic problems	32	22.6 (5.1)	16	20.4 (3.8)	15	18.9 (4.7)
Partner relationship problems	25	3.7 (3.2)	15	2.9 (3.3)	13	3.3 (3.1)
Parenting stress	33	88.6 (25.4)	19	77.1 (20.6)	17	80.2 (28.2)

**Table 3 ijerph-19-14693-t003:** Confidence intervals and effect sizes of change between pre-test and post-test and between pre-test and follow-up.

Outcome Variable	Pre-Test–Post-Test			Pre-Test–Follow-Up		
	*n*	95% CI	*d*	*n*	95% CI	*d*
Primary outcomes						
Internalizing psychopathology	20	0.98–7.97	0.60	20	0.66–5.74	0.59
Externalizing psychopathology	20	1.81–8.19	0.73	20	1.66–7.84	0.72
Secondary outcomes and mechanisms						
Attention problems	20	0.10–3.50	0.49	20	0.22–3.98	0.52
Executive function problems	18	3.85–20.81	0.72	18	1.17–18.33	0.57
Family systemic problems	16	0.54–5.58	0.65	15	0.18–6.75	0.58
Partner relationship problems	15	0.14–2.05	0.64	12	−0.05–1.30	0.59
Parenting stress	19	6.48–21.72	0.89	17	0.13–22.98	0.52

The 95% CI of mean difference between pre-test and post-test and between pre-test and follow-up questionnaires; *d* = Cohen’s *d* (within-group) effect size of change.

**Table 4 ijerph-19-14693-t004:** Correlations between all outcome measures.

	1.	2.	3.	4.	5.	6.	7.
Primary outcomes							
1. Internalizing psychopathology	-	0.30	0.15	0.26	0.23	0.32	0.39
2. Externalizing psychopathology	-	-	0.27	0.55	0.34	0.28	0.75
Secondary outcomes and mechanisms							
3. Attention problems	-	-	-	0.39	0.16	0.08	0.28
4. Executive function problems	-	-	-	-	−0.05	0.19	0.60
5. Family systemic problems	-	-	-	-	-	0.37	0.39
6. Partner relationship problems	-	-	-	-	-	-	0.39
7. Parenting stress	-	-	-	-	-	-	-

**Table 5 ijerph-19-14693-t005:** Correlations between difference scores (post-test minus pre-test) of all outcome measures.

	1.	2.	3.	4.	5.	6.	7.
Primary outcomes							
1. Internalizing psychopathology	-	0.62	0.68	0.38	0.37	0.03	0.52
2. Externalizing psychopathology	-	-	0.49	0.69	0.28	0.09	0.53
Secondary outcomes and mechanisms							
3. Attention problems	-	-	-	0.40	0.52	0.32	0.66
4. Executive function problems	-	-	-	-	0.35	0.04	0.69
5. Family systemic problems	-	-	-	-	-	0.63	0.54
6. Partner relationship	-	-	-	-	-	-	0.23
7. Parenting stress	-	-	-	-	-	-	-

## Data Availability

The data presented in this study are available on request from the corresponding author. The data are not publicly available due to ethical reasons (personal data).

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
