# Peer review of "BOAM: A Visual, Explanatory Diagnostic and Psychoeducation System Used in Collaboration with Families—Feasibility and Acceptability for Children Who Are Non-Responsive to Treatment as Usual"

_ijerph, 2022, doi:10.3390/ijerph192214693_

Round 1

Reviewer 1 Report

The manuscript is well-written, especially the Introduction part but as a reader it would have helped more if the results were also presented in pictorial/bar plot besides tables or paragraph .

The study has two major limitations; higher research dropouts and absence of control groups. Although authors have clearly mentioned these (and more) limitations in the Discussion, however it does weaken the claims of study. 

But I do think it is an important study (even with its limitations) and should considered for publication.

Author Response

Reviewer 1:

The manuscript is well-written, especially the Introduction part but as a reader it would have helped more if the results were also presented in pictorial/bar plot besides tables or paragraph.

The study has two major limitations; higher research dropouts and absence of control groups. Although authors have clearly mentioned these (and more) limitations in the Discussion, however it does weaken the claims of study. 

But I do think it is an important study (even with its limitations) and should considered for publication.

-> Thank you for your constructive feedback. You are right about the limitations of the study. Because of your feedback and the recommendations of other reviewers and the editor regarding these limitations, we decided to present our study as a feasibility and acceptability study, and put less emphasis on the results. Although we agree that in the first version of our manuscript, pictorial/bar plots would have been a good addition, we assumed that in the new version of our manuscript this would fit less well.

Reviewer 2 Report

The content

The article presents an alternative diagnostic and psychoeducation system designed for the families of children and youth with emotional, behavioral, or psychiatric challenges who were hard to treat with standard tools. The authors see these challenges occurring not only within the child but in a family setting, thus, requiring not to treat the child alone but afford parents a significant part in consideration and intervention. The suggested system named BOAM (Basic needs, Order, Autonomy, Meaning) harnesses parents' awareness and everyday experience to support the parenthood of the child with a challenge. 

Various sets of visual aids and images are supplied during the process to locate the child and challenges: their neurodevelopmental and emotional roots and the way they echo and grow during everyday life.

The model has been applied to 34 children diagnosed with various challenges. The children's parents, from different backgrounds, filled out a pre-test and a post-test before and after the intervention. The research results are presented in this article, followed with a discussion and conclusion on the system's efficacy. 

Developing this system seems such a clever and necessary idea. Parents do need to learn more equally in regard to their children's challenges and the way they challenge the family. The research idea is even more clever, and overall this article is necessary and original.

Some remarks

1. Accessibility of the visual tool: The imagery used with the parents seems valuable, and the illustrations are highly aesthetic. Still, I would see that they are even more accessible graphically and language-wise. More careful choices of graphics, color schemes, and words can make the visual tool accessible to even more parents.

2. The intro: The introduction combines the literature review with a lengthy explanation of BOAM. Would the authors consider dividing it into two? A literature review, followed by a separate section describing the BOAM system. I would consider shortening the latter, overshadowing the following parts: the research, discussion, and conclusions.

The research: To improve the clarity and strengthen the part of the research, would the authors consider the following?

  1. Move paragraph 1.7 into the "research section.
  2. See that paragraphs 1.7, 2.2, and 2.3 don't repeat what was explained previously during the background section and that the background material contains all the necessary information. 
  3. The headline "Materials and methods" on page 20 doesn't describe any materials. Would you reconsider the name?
  4. Would you consider improving the part on the research section even more? See that its parts have more order and the description more focused, the recruiting is explained thoroughly, and the Ethics explained with more detail on separate section.
  5. There is a heavy load of verbal information that is latter described in tables. Could you please see what is necessary and what is unnecessarily repeated?
  6. Line 73 (given de genetic...). Is it a mistake?

 It has been an honor to review this interesting article. 

Thank you,

Efrat

Author Response

Reviewer 2

The content

The article presents an alternative diagnostic and psychoeducation system designed for the families of children and youth with emotional, behavioral, or psychiatric challenges who were hard to treat with standard tools. The authors see these challenges occurring not only within the child but in a family setting, thus, requiring not to treat the child alone but afford parents a significant part in consideration and intervention. The suggested system named BOAM (Basic needs, Order, Autonomy, Meaning) harnesses parents' awareness and everyday experience to support the parenthood of the child with a challenge. 

Various sets of visual aids and images are supplied during the process to locate the child and challenges: their neurodevelopmental and emotional roots and the way they echo and grow during everyday life.

The model has been applied to 34 children diagnosed with various challenges. The children's parents, from different backgrounds, filled out a pre-test and a post-test before and after the intervention. The research results are presented in this article, followed with a discussion and conclusion on the system's efficacy. 

Developing this system seems such a clever and necessary idea. Parents do need to learn more equally in regard to their children's challenges and the way they challenge the family. The research idea is even more clever, and overall this article is necessary and original.

-> Thank you for reading our manuscript so carefully, and for your compliments.

Some remarks

  1. Accessibility of the visual tool: The imagery used with the parents seems valuable, and the illustrations are highly aesthetic. Still, I would see that they are even more accessible graphically and language-wise. More careful choices of graphics, color schemes, and words can make the visual tool accessible to even more parents.

-> The development of the models (by second author DT) has been going on for years and still is an ongoing process. From the start, the models have been used and evaluated in practice. Both the content and the form, as well as the presentation of the models and the development of the terminology have come about in this way. We are very open for feedback. However, on the basis of this comment, we are not sure what the reviewer thinks could be improved specifically. If possible, that reviewer may give us more specific feedback in the future.

  1. The intro: The introduction combines the literature review with a lengthy explanation of BOAM. Would the authors consider dividing it into two? A literature review, followed by a separate section describing the BOAM system. I would consider shortening the latter, overshadowing the following parts: the research, discussion, and conclusions.

-> Thank you for your suggestion. We now combined the first few paragraphs of the introduction in a ‘literature review’, which we also used as a subheading (level 2 heading). For the explanation about the BOAM diagnostic process, we did not only use a Subheading, but also Level 3 subheadings, to separate the process in different steps. We hope this improves the structure/order of the introduction. After consideration, we decided not to shorten the text about the BOAM diagnostic process for several reasons. The first is associated with the change in aims of the study: the aim is no longer to study the effectiveness and working mechanisms of the BOAM diagnostic trajectory and subsequent treatment, but to study the feasibility and acceptability of BOAM and the feasibility of outcome measures. This means that there is less emphasis on the research part of the article, and there is more space for an introduction of the BOAM process. In order to get a true understanding of what BOAM entails, we thought that it was important to give a lot of information about the different diagnostic steps.

The research: To improve the clarity and strengthen the part of the research, would the authors consider the following?

  1. Move paragraph 1.7 into the "research section.

-> As explained above, we now changed the aims of the current study, therefore paragraph 1.7 was also adjusted. We do think it is important to introduce the research aims at the end of the introduction and before the methods section.

  1. See that paragraphs 1.7, 2.2, and 2.3 don't repeat what was explained previously during the background section and that the background material contains all the necessary information. 

-> We have checked the document for repetitions. Some repetitions seemed inevitable, because thought it was important to make explicit how the information from the diagnostic trajectory (Introduction) translates to the treatment phase (Intervention subsection).

  1. The headline "Materials and methods" on page 20 doesn't describe any materials. Would you reconsider the name?

-> Yes you are right. The heading Methods seems more suitable for our study than ‘Materials and methods’. We thought however, that the heading ‘Materials and methods’ was prescribed by the journal. We now adjusted the heading to ‘Methods’ but can of course change it back if the journals prefers this.

  1. Would you consider improving the part on the research section even more? See that its parts have more order and the description more focused, the recruiting is explained thoroughly, and the Ethics explained with more detail on separate section.

-> We added the following information to the Procedure subsection: “Children and their parents either admitted themselves for the BOAM trajectory, or were referred to the BOAM trajectory by their general practitioner or mental health professional. If they had prior treatment as usual, to which they did not respond, or after which they relapsed, they could be included in the current study. Parents and children were asked to participate in the current study, but they could also do the BOAM diagnostic trajectory if they decided not to partake in the study.”

  1. There is a heavy load of verbal information that is latter described in tables. Could you please see what is necessary and what is unnecessarily repeated?

-> Thank you for pointing this out to us. We think that this is no longer a problem now that the statistical tests have been removed from the manuscript.

  1. Line 73 (given de genetic...). Is it a mistake?

-> Yes, this was a mistake, thank you for reading so attentively. We adjusted ‘de’ to ‘the’.

It has been an honor to review this interesting article. 

-> Thank you again!

Reviewer 3 Report

 Dear Authors

Derived from the review of the manuscript, I consider that in general the document is quite well written, it has a robust theoretical basis that allows arguing the development of the BOAM diagnostic model as an alternative for the treatment of psychopathology in children and adolescents considering the integration of the family as an element that facilitates and improves the prognosis of treatment.

Likewise, it is considered that the methodological approach is robust and the analytical approach through a multilevel model is appropriately justified, however, the sample size can affect the estimation of the parameters of a model of this nature.

Continuing with the theoretical assumptions that a multilevel model must meet, it would be convenient if the authors could also include the evidence indicating that the dependent variables  assumed the following assumptions: 1) homogeneity of variances, 2) normality of residuals. The presentation of this information provides certainty about the validity of the statistical conclusion presented in the study.

I noted that most of the validity threats to the study are commented on in the discussion, with which I agree.

I hope that these brief comments help to strengthen your manuscript.

Kind regards

Author Response

Reviewer 3

Derived from the review of the manuscript, I consider that in general the document is quite well written, it has a robust theoretical basis that allows arguing the development of the BOAM diagnostic model as an alternative for the treatment of psychopathology in children and adolescents considering the integration of the family as an element that facilitates and improves the prognosis of treatment.

Likewise, it is considered that the methodological approach is robust and the analytical approach through a multilevel model is appropriately justified, however, the sample size can affect the estimation of the parameters of a model of this nature.

Continuing with the theoretical assumptions that a multilevel model must meet, it would be convenient if the authors could also include the evidence indicating that the dependent variables  assumed the following assumptions: 1) homogeneity of variances, 2) normality of residuals. The presentation of this information provides certainty about the validity of the statistical conclusion presented in the study.

I noted that most of the validity threats to the study are commented on in the discussion, with which I agree.

I hope that these brief comments help to strengthen your manuscript.

-> Thank you very much for your valuable feedback and suggestions. We agree that it would have been good to check the assumptions more thoroughly by checking the homogeneity of variance and normality of residuals. However, on the basis of the comments of other reviewers and the editor, we decided to remove the statistical tests from the study.

Reviewer 4 Report

I enjoyed this article, I found it very interesting, both for the innovation of the topic, and for the complexity of the contents and the clarity of presentation.

I found the BOAM model very interesting. I confess that I found the images fascinating and full of symbolism and I think they can catch the attention of children / adolescents and parents.

Based on my experience as a clinical psychologist, I believe that the BOAM model can facilitate the understanding of the causes of difficulties, in a systemic vision not only for specialists but above all for patients and their families, improving their empowerment and participation in diagnostic and therapeutic process. I believe it is useful in guiding the therapeutic path and in facilitating psychoeducation for minors and their families.

Regarding the critical issues, I invite the authors to explain the treatment in more detail (paragraph 2.3): reference is made to the "steps" but not to the type of treatment used.

I believe that a possible future study in which minors are subjected to pre-intervention and post-intervention evaluation (with neuropsychological and cognitive tests) would allow us to evaluate whether there has actually been an improvement with respect to the difficulties of attention and executive functions, since the only perception of the parent can underestimate or overestimate the problem.

Author Response

Reviewer 4

I enjoyed this article, I found it very interesting, both for the innovation of the topic, and for the complexity of the contents and the clarity of presentation.

I found the BOAM model very interesting. I confess that I found the images fascinating and full of symbolism and I think they can catch the attention of children / adolescents and parents.

Based on my experience as a clinical psychologist, I believe that the BOAM model can facilitate the understanding of the causes of difficulties, in a systemic vision not only for specialists but above all for patients and their families, improving their empowerment and participation in diagnostic and therapeutic process. I believe it is useful in guiding the therapeutic path and in facilitating psychoeducation for minors and their families.

-> It was a pleasure to read that our manuscript was not only reviewed by scientists but also by a clinical psychologist, and that your impression was that the models would be useful in clinical practice. Thank you for not only reading our manuscript but also studying the BOAM models carefully.

Regarding the critical issues, I invite the authors to explain the treatment in more detail (paragraph 2.3): reference is made to the "steps" but not to the type of treatment used.

-> Thank you for your suggestion. We added the following information to the Intervention subsection: “Subsequent treatment is not protocolized but dependent on the results of the diagnostic trajectory and the needs of the family. The type of treatment can be described as a combination of parental guidance, cognitive behavioural therapy for the child, and family therapy. In that sense, the treatment phase is similar to treatment as usual. Our clinical impression is that there are two important differences between a treatment after a BOAM diagnostic trajectory and treatment as usual. The first is that because both parents and children have a greater understanding of the problems after the diagnostic trajectory, they also understand better what is needed in terms of treatment, they may be better motivated and more compliant with the current treatment. Therefore, they may benefit more from treatment that may not be very different than the treatment they have received earlier. The second is that the BOAM models help to stay more attuned to understanding the basic needs of the child (for example of safety and predictability), the child and parents learn how to meet these basic needs, instead of treatment that is focused on meeting certain expectations when the basic needs are not met. The third is that that relatively more time is spent on supporting the ordering process, also in children who do not have an ASD or ADHD classification. The fourth is that parents are stimulated to also apply the models on themselves, and for example gain more understanding on what they themselves need to be able to fulfil the needs of the child while not becoming exhausted or overwhelmed themselves. The fifth is that the visual help-model is used, which is explained in more detail in the next paragraph. The help-model gives support and structure in ‘crisis-situations’, by giving parents steps to follow in these situations.”

The information about the use of the ‘help-model’ in the Intervention subsection has been expanded: “use the BOAM help model (see Figure 4) in case of child emotional or behavioral difficulties, the so-called ‘dragon-reactions’ as explained in paragraph 1.3. Step 1 entails giving the child recognition, and especially recognizing the ‘primary problem’. In this step, parents are taught to make use of techniques that are based on non-violent resistance in families, as developed by Omer [51], In step 3, the parents can support the child by fulfilling the core needs, which may be one of the prior described steps (but focused on the specific situation that the problem arose in), or can exist of giving the child information (for example by putting the event in the context that the child may not have been able to recognize), or by offering cognitive restructuring).”

I believe that a possible future study in which minors are subjected to pre-intervention and post-intervention evaluation (with neuropsychological and cognitive tests) would allow us to evaluate whether there has actually been an improvement with respect to the difficulties of attention and executive functions, since the only perception of the parent can underestimate or overestimate the problem.

-> That is a great idea. We have added the word neuropsychological tests to the following sentence in the Discussion: “Future research could use a methodologically improved design, such as a (quasi) experimental design, differentiate between the effects of the BOAM diagnostic trajectory and subsequent treatment, and use more diverse informants or more objective measures such as observations and neuropsychological tests.”
We also added the following sentence: “This is needed to evaluate whether there has actually been an improvement with respect to the intrapersonal and interpersonal factors, and whether these can be attributed to treatment that is based on the BOAM diagnostics.”

Reviewer 5 Report

As the authors point out in limitations, the study has serious methodological problems. The main one is not having a control group, so the results found cannot be attributed to the BOAM system. But there are more issues in this regard that prevent the publication of the article in a high-impact journal:

-The sample size required for the study has not been calculated. The sample is very small for the large number of variables that are evaluated and there is also a very high percentage of missing values. This discourages multivariate analysis and casts doubt on the reliability of the results.

-The sample is very heterogeneous with respect to diagnoses and is made up mainly (especially after dropouts) of parents with a high educational level. This calls into question the representativeness of the sample and the generalizability of the results.

-Given the heterogeneity of diagnoses, it is not clear what treatment as usual consists of, who is considered a non-responder, and how relapse is defined.

-In the introduction, in the sections in which the diagnostic process is described, many affirmations are made without supporting them with bibliographic citations.

-When it comes to treatment, the strategies used do not seem to differ from treatment as usual (see lines 446-452).

-A non-validated family dynamics questionnaire is used.

Author Response

Reviewer 5

As the authors point out in limitations, the study has serious methodological problems. The main one is not having a control group, so the results found cannot be attributed to the BOAM system. But there are more issues in this regard that prevent the publication of the article in a high-impact journal:

-> Thank you for critically reading our manuscript and for your very right feedback.

-The sample size required for the study has not been calculated. The sample is very small for the large number of variables that are evaluated and there is also a very high percentage of missing values. This discourages multivariate analysis and casts doubt on the reliability of the results.

-> We agree with your point of feedback. The editor suggested to remove the statistical tests, which would solve this issue. We indeed decided to do just that, and changed to aim of the study so that it is now a feasibility and acceptability study.

-The sample is very heterogeneous with respect to diagnoses and is made up mainly (especially after dropouts) of parents with a high educational level. This calls into question the representativeness of the sample and the generalizability of the results.

-> This is indeed another limitation of the study. Due to the decision to change the aim of the study, we think that the consequences of this limitation are less serious than in the last version of our manuscript. However, we did include this issue more explicitly in the limitations section: “Another limitation was the high percentage of missing measurements at both post-test (41%) and follow-up (41%). In our study sample, there was a high percentage of parents with a high level of education, and of the parents who dropped out of the research, slightly more parents had a lower level of education than the parents who did complete the post-test and/or follow-up measurement. It is unknown whether the results of this study are generalizable. Also, we cannot rule out the possibility that parents who dropped out of the research would evaluate the BOAM trajectory less positively than the parents who did complete the post-test or follow-up measurement.”

-Given the heterogeneity of diagnoses, it is not clear what treatment as usual consists of, who is considered a non-responder, and how relapse is defined.

-> Indeed the research group was quite heterogeneous. In this first study, we decided to especially ‘follow’ clinical practice instead of setting up a strict research protocol, which brought about serious methodological limitations, and it is not possible to draw clear conclusions about the effectiveness of BOAM. Therefore, we completely agree that study should be named a feasibility study.

-In the introduction, in the sections in which the diagnostic process is described, many affirmations are made without supporting them with bibliographic citations.

-> We added 15 bibliographic citations to the introduction, of which 10 to the sections in which the diagnostic process is described. Also, we added the words ‘according to BOAM theory’ a few times, if no bibliographical citation was available.

-When it comes to treatment, the strategies used do not seem to differ from treatment as usual (see lines 446-452).

-> Thank you for your comment. We tried to address this issue and make the differences more clear by adding the following text to the Intervention subsection: “Subsequent treatment is not protocolized but dependent on the results of the diagnostic trajectory and the needs of the family. The type of treatment can be described as a combination of parental guidance, cognitive behavioural therapy for the child, and family therapy. In that sense, the treatment phase is similar to treatment as usual. Our clinical impression is that there are two important differences between a treatment after a BOAM diagnostic trajectory and treatment as usual. The first is that because both parents and children have a greater understanding of the problems after the diagnostic trajectory, they also understand better what is needed in terms of treatment, they may be better motivated and more compliant with the current treatment. Therefore, they may benefit more from treatment that may not be very different than the treatment they have received earlier. The second is that the BOAM models help to stay more attuned to understanding the basic needs of the child (for example of safety and predictability), the child and parents learn how to meet these basic needs, instead of treatment that is focused on meeting certain expectations when the basic needs are not met. The third is that that relatively more time is spent on supporting the ordering process, also in children who do not have an ASD or ADHD classification. The fourth is that parents are stimulated to also apply the models on themselves, and for example gain more understanding on what they themselves need to be able to fulfil the needs of the child while not becoming exhausted or overwhelmed themselves. The fifth is that the visual help-model is used, which is explained in more detail in the next paragraph. The help-model gives support and structure in ‘crisis-situations’, by giving parents steps to follow in these situations.”

-A non-validated family dynamics questionnaire is used.

-> We wanted to measure family system problems, but were also in doubt whether we should or should not report the results of this questionnaire, and we still have some doubts about it. If the editor prefers us to remove this questionnaire from the study, we will do that.  

Reviewer 6 Report

Thank you for the opportunity to review the manuscript titled "BOAM: a visual, explanatory diagnostic and psychoeducation system used in collaboration with families; Effects on children who were non-respondent to treatment as usual".

The manuscript taps an important topic concerning an important population. It introduces an original and significant contribution to the literature by using a robust design to test a new diagnostic system that explains symptomology and related risk factors to children who are non-respondent to treatment in a simple way. The authors hypothesized that this approach would have positive short- and long-term impacts on both children and their parents.

While the study has several strengths and potentials, I strongly recommend that authors consider their research as a pilot study. With the small sample size, lack of a control group, and lack of collateral reports, there are several critical threats to the internal and external validity of the findings.

 Therefore, the title, aims, discussion, and conclusions should clearly reflect that this is a small scale investigation.

I suggest the purpose to be to evaluate the feasibility and preliminary efficacy of BOAM…

Thank you

Author Response

Reviewer 6

Thank you for the opportunity to review the manuscript titled "BOAM: a visual, explanatory diagnostic and psychoeducation system used in collaboration with families; Effects on children who were non-respondent to treatment as usual".

The manuscript taps an important topic concerning an important population. It introduces an original and significant contribution to the literature by using a robust design to test a new diagnostic system that explains symptomology and related risk factors to children who are non-respondent to treatment in a simple way. The authors hypothesized that this approach would have positive short- and long-term impacts on both children and their parents.

-> Thank you for reading our manuscript and for your very clear recommendations, that have helped us to improve our manuscript.

While the study has several strengths and potentials, I strongly recommend that authors consider their research as a pilot study. With the small sample size, lack of a control group, and lack of collateral reports, there are several critical threats to the internal and external validity of the findings.

-> Thank you for your suggestion. You are very right that our study is more a pilot study than an effectiveness study. The editor gave us another suggestion, which was to consider our study as a feasibility study, as you also did in your last comment. We decided to follow that suggestion.

Therefore, the title, aims, discussion, and conclusions should clearly reflect that this is a small scale investigation.

-> We indeed have made the changes to all of the sections of the manuscript, in order to better reflect the small scale investigation.

I suggest the purpose to be to evaluate the feasibility and preliminary efficacy of BOAM…

-> Thank you for your right suggestion. We have adjusted the title of the manuscript to “BOAM: a visual, explanatory diagnostic and psychoeducation system used in collaboration with families; Feasibility and acceptability for children who were non-respondent to treatment as usual”. The aim was adjusted to: “In the current study, we evaluate the feasibility and acceptability of the BOAM diagnostic models in a diagnostic trajectory and subsequent treatment that is informed by this trajectory, in children and adolescents who were non respondent to, or relapsed after treatment as usual. As a preparation for future effectiveness studies, we assessed the feasibility of questionnaires measuring the most important primary outcomes (child/adolescent internalizing and externalizing psychopathology) of the BOAM trajectory, and possible secondary outcomes and working mechanisms (attention problems, executive functioning problems, family system problems, partner relationship and parenting stress).”

Reviewer 7 Report

Thank you for the opportunity to review this paper.

There was not much for me to add as the novelty and soundness of this contribution is of a high standard, with a much-needed topic for discussion. Especially at a time (pandemic) where mental health issues are more prominent than ever, it was refreshing to read the authors’ study into children’s mental health. I was particularly interested in the scientific background and the authors’ discussion of findings.

As social and emotional development includes such a wide array of sub-milestones, a discussion on the key aspects of these two developmental domains could have been discussed further. The discussion on previous studies/literature was also well noted. However, the long-term temporal trends and geographical trends in children’s mental health could have been explored further.

Lastly, the paper has potential to make the contribution of this research, to extend beyond policy-level and for teacher practice. Thus, the authors may consider providing further recommendations/suggestions of their findings for the immediate educational and/or care environment and suggestions to support the wider cohort.

Author Response

Reviewer 7

Thank you for the opportunity to review this paper.

There was not much for me to add as the novelty and soundness of this contribution is of a high standard, with a much-needed topic for discussion. Especially at a time (pandemic) where mental health issues are more prominent than ever, it was refreshing to read the authors’ study into children’s mental health. I was particularly interested in the scientific background and the authors’ discussion of findings.

-> Thank you for you positive and encouraging feedback.

As social and emotional development includes such a wide array of sub-milestones, a discussion on the key aspects of these two developmental domains could have been discussed further. The discussion on previous studies/literature was also well noted. However, the long-term temporal trends and geographical trends in children’s mental health could have been explored further.

-> Thank you for your suggestion. We have added the following text to the discussion on : “BOAM theory suggests that characteristics of modern society are risk factors for a lack in fulfilment in the need for order, and thus for the development of social-emotion or behavioral problems. The environment that children grow up in may be qualitatively good, but quantitatively overstimulating and overcharging, due to an overload in sensory stimuli, a high complexity, and relatively high expectations and demands on the areas of social and executive functioning, given the neurodevelopmental make-up of an individual child. Attuning of the environment to the needs of the individual child is important, as children also need proper stimulation and rich environments [66], and may also benefit from positive expectations of parents for example [67]. Research has shown that overstimulation in the form of an excessive amount of time of watching rapidly sequenced television in young children is associated with later attention problems [68,69]. Although the authors are not aware of research addressing the hypothesis growing up in complex environments is a risk factor for the development of social-emotional or behaviour problems, research has been done on the association between living in an urban environment and mental health in children. The percentage of children growing up in urban living environments has been growing in the past decades [70]. Also, the prevalence of mental health disorders, such as autism, depression and anxiety, have increased [71,72]. Both autism and common mental disorders have been shown to be related to urbanicity in many different countries [73,74]. A meta-analysis on rural-urban migration and mental health in Chinese migrant children showed that compared with urban children, migrant children had more mental health problems [75]. One of the explanations that the authors gave for this difference was that mobile life and the fast lifestyle may contribute to psychological discomfort of migrant children [75]. Future research may not only assess the proper amount and quality of stimulation richness of the environment and expectations for children in general, but may also look at the right attunement of these elements to the neurodevelopmental capacities of the child.

The BOAM models bring awareness to the possibility that children with social-emotional or behavioural problems suffer from ordering problems that are related to being overdemanded and overstimulated. Possible overstimulation or overdemanding is oftentimes attended to in treatment as usual for children with ASD, but is oftentimes overlooked in the treatment of children with other classifications, while it has been shown that other psychological problems are also associated with sensory processing difficulties and other neuropsychological problems [11,33]. Although ASD is diagnosed on the basis of behavioural criteria, even milder forms of ASD are oftentimes seen as a lifelong sensory dysfunction and disorder of neurodevelopment, which implies neural dysfunction in the child [76]. From the BOAM perspective, similar behavioural characteristics may be seen as a combination of a neural sensitivity, and a mismatching environment, and are called ordering problems. Therefore, BOAM theory offers a more hopeful perspective with regard to ordering problems and associated social and/or executive functioning problems by revealing the choice to adjust the environment better to the needs of the child and give it more time to develop in order to avoid psychological experiences of failure and its consequences for the development. The models do not only identify possible problems areas, but also show the possibilities for, and the right conditions for healthy development.”

Lastly, the paper has potential to make the contribution of this research, to extend beyond policy-level and for teacher practice. Thus, the authors may consider providing further recommendations/suggestions of their findings for the immediate educational and/or care environment and suggestions to support the wider cohort.

-> The following text has been added to the Discussion: “The BOAM models aim to depict universal processes that are related to psychological functioning. They are not meant to be used in the context of mental health care only but may also have implications for prevention programs, and for educational and care environments. What may be needed is a heightened awareness that for example activities such as school camps or extracurricular art classes may be positively stimulating for one child, but may be too unpredictable, overstimulating or overwhelming for another child, thereby undermining the feeling of basic security for this child. Knowledge about the negative effects of avoidance on anxiety seems to be widespread, which may stimulate the opinion that children learn to deal with difficult situations, such as new situations of overstimulating situations by coming into contact with them. Although this may be true, on the basis of BOAM theory, enough attention should be given to find the right balance, and to give the child the proper support, preparation, and structure in these situations. In a report on early development and learning, emphasis was not only placed on ways in which children learn new skills and competencies, but also on the importance of secure relationships in learning (and thus not too many changes in people involved), and the negative effect of chronic high stress on learning [66]. That means that if a child experiences high stress or does not feel secure, this should be attended to first, and this stimulation and expectations should be adjusted for this child.”

Round 2

Reviewer 5 Report

The authors have improved several of the suggested aspects. Converting the study into one of feasibility and acceptability fits better with the methodology used. Although the article has improved, I still consider that the methodology and the type of study would fit better in a journal with less impact, while in this journal the future study on effectiveness proposed by the authors would be publishable.

Author Response

-> Thank you for taking the time to review our manuscript another time, and recognizing the improvement. We understand your comment regarding the methodology and type of study.